# The Multimodality Management of Malignant Peripheral Nerve Sheath Tumours

**DOI:** 10.3390/cancers16193266

**Published:** 2024-09-26

**Authors:** Remus Seres, Hassan Hameed, Martin G. McCabe, David Russell, Alexander T. J. Lee

**Affiliations:** 1Department of Medical Oncology, The Christie NHS Foundation Trust, Manchester M20 4BX, UK; 2Division of Cancer Sciences, Faculty of Biology, Medicine and Health, University of Manchester, Manchester M13 9PL, UK; 3Department of Radiology, Lancashire Teaching Hospitals NHS Trust, Chorley PR7 1PP, UK; 4NHS England Highly Specialised Service for Complex Neurofibromatosis Type 1: Manchester, Manchester Centre for Genomic Medicine, St Mary’s Hospital, Manchester M13 9WL, UK

**Keywords:** malignant peripheral nerve sheath tumour, neurofibromatosis type 1, nerve, neurofibroma, molecular alterations, multimodality management

## Abstract

**Simple Summary:**

The landscape of malignant peripheral nerve sheath tumours (MPNSTs) is usually challenging both in terms of recognition and management. Despite a low incidence in the general population (0.001%), MPNST is an important cause of mortality in the neurofibromatosis type 1 (NF1) population. It is essential for a multi-disciplinary collaboration to achieve the best possible outcome. The aim of our paper was to contribute with a comprehensive review from the literature of the best multi-modality ways that show improvements in terms of survival and address potential future treatment approaches based on the molecular alterations seen in these tumours.

**Abstract:**

Malignant peripheral nerve sheath tumours (MPNST) are aggressive sarcomas that have nerve sheath differentiation and can present at any anatomical site. They can arise from precursor neurofibroma in the context of neurofibromatosis type 1 (NF1) or as de novo and sporadic tumours in the absence of an underlying genetic predisposition. The primary therapeutic approach is most often radical surgery, with non-surgical modalities playing an important role, especially in locally advanced or metastatic cases. The aim of multimodality approaches is to optimize both local and systemic control while keeping to a minimum acute and late treatment morbidity. Advances in the understanding of the underlying biology of MPNSTs in both sporadic and NF-1-related contexts are essential for the management and implementation of novel therapeutic approaches.

## 1. Introduction

MPNST is a rare subtype of soft tissue sarcoma that typically exhibits aggressive clinical behaviour. MPNST has an estimated incidence of approximately 1.5 cases per million per year in the general population. However, up to 50% of all MPNSTs occur in individuals with NF1, where the reported lifetime risk of developing these tumours is 8–13% [1,2]. MPNST can arise from normal nerves or precursor neurofibromas (the hallmark benign nerve sheath tumour of NF1) but not from Schwannomas (the benign nerve sheath tumours that arise from Schwann cells and are characteristic of neurofibromatosis type 2). Up to 10% of MPNSTs develop as radiation-associated secondary malignancies, arising within or adjacent to previous radiotherapy fields many years after prior irradiation [3].

MPNST remains a leading cause of excess and early mortality in NF1 despite earlier access to diagnosis, better surgical approaches and updates of the histological and molecular classification.

Survival outcomes have historically been regarded as worse in NF1-related MPNST (5 year survival rates 16–26%) compared to sporadic cases (42–55%), although a focus on more recent series suggests this difference may be less than previously suspected [4].

While the histological diagnosis of sporadic MPNST can be challenging, the appearance of overt malignant features within a precursor plexiform neurofibroma (PN) readily achieves the diagnosis of NF1-associated MPNST. More challenging can be the distinction between neurofibroma with premalignant changes or low-grade malignancies [5,6].

## 2. Clinical Presentation and Diagnosis

### 2.1. Clinical Features

MPNST typically presents as a growing soft tissue mass with or without associated pain. In NF1, distinguishing between benign and malignant lesions can be clinically challenging; rapid growth, an increase in pain and/or new neurological symptoms associated with neurofibromas should arouse suspicion [7].

However, in both sporadic and NF1-associated MPNST, tumours can arise in the absence of an overt precursor lesion.

MPNSTs tend to present at an earlier age in patients with NF1 (third or fourth decade) as compared to sporadic cases (seventh decade) [2]. The most common sites of anatomical origin are the extremities and trunk, followed by the head and neck. Less frequently (up to 3%), it is located in the retroperitoneal region, making them the fifth most common sarcoma subtype in that anatomical area [8,9,10].

### 2.2. Imaging Characteristics

Imagining studies are essential to characterize the tumour extent and to differentiate between MPNST and benign nerve sheath neoplasms. Magnetic resonance imaging (MRI) is generally the preferred modality for assessment of the primary lesion, where features potentially indicative of malignancy include lesion size >5 cm, surrounding peritumoral oedema, intra-tumoral heterogenous enhancement on T1-T2 weighted images and irregular and/or invasive margins (Figure 1A,B) [11,12,13,14,15].

MPNSTs typically have increased metabolic activity compared to benign neurofibroma. The use of fluorodeoxyglucose (18F-FDG) positron emission tomography (PET) provides additional means of distinguishing between malignant and non-malignant nerve sheath tumours. FDG-PET-CT is reported to have a sensitivity and specificity of over 90% for diagnosing MPNST in patients with symptomatic lesions, often by applying a cut-off in maximum standardized uptake value (SUV max) between 3 and 4 (Figure 1C). These results are variably limited by factors that include (a) dichotomization between malignant and non-malignant lesions that fails to account for the gradation of malignant progression, (b) a change in the histological classification of benign, atypical and malignant nerve sheath tumours subsequent to the majority of reported studies, (c) inclusion of a few atypical neurofibromas (and thus provide limited information in the use of FDG-PET to identify premalignant disease) and (d) inconsistency in criteria used to define PET positive or negative lesions [16,17,18,19].

For confirmed or suspected malignant lesions, staging imaging is required to assess for the presence and extent of distant spread. MPNSTs usually disseminate via perineural invasion or haematogenous routes. The lungs are the most common metastatic site, with spread to the liver, bone, brain and/or soft tissues relatively uncommon. Lymph node metastases are quite rare, appearing in less than 10% of the cases. As such, computerized tomography (CT) imaging of the chest is generally adequate to assess for metastatic disease in the absence of clinical indication of extrapulmonary metastasis.

MPNSTs are staged according to the American Joint Committee on Cancer (AJCC) tumour, node and metastasis (TNM) system, along with other soft tissue sarcomas of the trunk and extremities [20].

### 2.3. Histology and Molecular Pathology

A clear and confident diagnosis of MPNST requires histopathological assessment, and so a biopsy of any suspicious soft tissue growth is typically indicated. In selected cases with very high clinical and radiological suspicion, surgery without prior biopsy may be preferred. In NF1, needle biopsies of suspect lesions are prone to false negative findings because of intralesional spatial heterogeneity and interposed areas of benign, pre-malignant and malignant change [21,22]. In such cases, the broader clinical context should be considered to avoid potential undertreatment. Targeting needle biopsies to areas of suspect lesions based on higher grade appearances on MRI and/or 18F-FDG-PET may improve diagnostic accuracy [23].

MPNST are typically firm and large tumours, containing areas of necrosis and haemorrhage [24]. Microscopically, there are specific elements that should be present, including high cellularity, palisade/rosette-like arrangements, asymmetric spindle cells and variable mitotic activity. The Federation Nationale des Centres de Lutte Contre le Cancer (FNCLCC) categorizes tumours as low-grade (FNCLCC 1) or high-grade (FNCLCC 2 or 3) on the basis of tumour differentiation, mitosis and necrosis. Both systems are prognostic, with high-grade predicting a poorer outcome [20].

There is no unique immunophenotypic marker for MPNST; therefore, differentiating it from other soft tissue sarcomas can be challenging. S100 is a marker of Schwann cell-derived tumours but can be absent in up to 40% of MPNST. SOX10 is a specific neural marker that likewise has limited sensitivity [25]. Nestin is an intermediate filament protein that is strongly expressed in the cytoplasm of MPNST, and it is more sensitive than other neural markers in the diagnosis of MPNST [26,27,28,29]. Loss of nuclear H3K27me3 is present in HG sporadic and radiation-induced tumours but lacks sensitivity in NF1-associated tumours [30]. Loss of INI1 expression is commonly seen in epithelioid MPNST [31,32]

At present, the diagnosis is mainly based on the exclusion of other mimicking tumours. Rare morphological variant subtypes include epithelioid and Triton (rhabdomyoblastic) MPNST. Triton tumours, in particular, are associated with aggressive clinical behaviour and poor outcome [33]. Desmin, myogenin and MyoD1 are present in rhabdomyosarcomatous elements.

Genomic study of preclinical models and clinical tissue samples from malignant and premalignant nerve sheath tumours has improved the characterization of the molecular alterations associated with the progression from benign neurofibroma to MPNST, with description of recurrent alterations in a number of key genes and molecular pathways as follows:

*NF1:* The key aspect of the molecular pathology in MPNST is related to the neurofibromatosis type 1 (*NF1)* gene on chromosome 17q11.2. *NF1* encodes neurofibromin, a GTPase tumour suppressor protein that negatively regulates RAS activation and downstream signalling pathway activity. Loss of heterozygosity (LOH) through partial or complete chromosomal deletion typically occurs in individuals with NF1 when the remaining allele is loss in tumour cells, leading to complete loss of neurofibromin function [34,35,36]. Somatic NF1 mutations have also been detected in more than 40% of sporadic MPNSTs, showing that NF1 inactivation plays an essential role in this tumour development [37]

*TP53:* Loss of p53 function contributes to genomic instability and tumour progression. Mutations or copy number alterations of *TP53*, a key tumour suppressor gene, are found in a significant proportion of MPNST but not in PN [38,39]. These translational observations, in addition to the ability of co-mutation of *TP53* with NF1 to result in MPMST development in mouse models, indicate that *TP53* inactivation contributes to the progression of neurofibroma and represents a substantial risk of MPNST transformation [40,41].

*CDKN2A/B:* Deletions or mutations in cyclin-dependent kinase inhibitors (*CDKN2A/B)*, which encode the cell cycle regulators p16 and p14ARF, are frequently observed in MPNSTs. The homozygous deletion of 9p21.3 containing *CDKN2A* in atypical neurofibroma indicates that this is an early step in the progression of benign neurofibroma to atypia. These alterations lead to dysregulation of the cell cycle and enhanced proliferation [42,43,44,45]. The *CDKN2A* mutation is considered an early mutation for MPNST.

*PRC2:* Mutations that impact upon poly-comb repressive complex 2 (PRC2), a protein complex implicated in epigenetic regulation, especially in its components SUZ12 or EED has been identified in a vast majority of MPNST across several studies. This epigenetic dysregulation through loss of PRC2 function plays an essential role in the development of MPNST. PRC2 mutations appear to trigger tumorigenesis by amplifying RAS pathway activation, an alteration that may have therapeutic potential [46,47,48].

*BRAF: BRAF* p(V600E) mutations are found in a subset of sporadic MPNST—the case report literature describes a single patient with MPNST with demonstrated BRAF V600E mutation treated effectively with the BRAF inhibitor vemurafenib [49]. The wider utility of targeting BRAF mutations in MPNST is otherwise unexplored.

Some evidence suggests that sporadic MPNST has similar mutant genes, but in a different order. For instance, the somatic *NF1* mutation of sporadic MPNST is similar to the NF1-associated one. *CDKN2A* or *PRC2* mutations have also been detected in sporadic MPNST. On the other hand, point mutations, such as BRAFV600E and NRAS Q61, are detected exclusively in sporadic cases [50,51].

## 3. Multimodality Management of MPNSTs

The optimal treatment approach may vary between individual cases of MPNST and should consider factors like size, tumour characteristics and location or stage at diagnosis. Given the interpatient variability and overall rarity of these cancers, case discussion and agreement of individualized management plans by specialist sarcoma multidisciplinary teams are recommended. In line with other sarcoma types, management in specialist sarcoma centres is associated with improved cancer outcomes [20].

## 4. Localized MPNST

For localized MPNST, complete surgical excision with clear margins is a pivotal element of potentially curative management. Generally, a R0 resection is recommended with at least 2 cm margin in all directions, but in MPNSTs this is not always achieved due to the nature of the anatomical position of these tumours [52,53]. Positive surgical margins have been repeatedly reported as associated with worse local control and overall survival rates in localized MPNST [54].

MPNST of the extremities have superior tumour control and R0 resection, likely contributing to better outcomes compared to truncal or head and neck tumours [55]. Current guidelines recommend limb-preserving surgery (LPS); nevertheless, up to 5–10% will undergo an amputation due to a more conservative approach not being feasible [55,56]. Combining radiotherapy with LPS has been shown to preserve functionality without impairing oncological outcomes. For extremity tumours not amendable to resection without significant morbidity or amputation, neo-adjuvant RT combined with/without isolated limb perfusion (ILP) with melphalan and tumour necrosis factor alpha, followed by resection, can contribute to successful limb preservation [57]. ILP, with or without adjuvant RT, was shown to have similar oncological outcomes to only adjuvant RT, but limb salvage rates (>80%) were reported to be superior with the addition of ILP [58,59]. In most centres, peri-operative RT is the common treatment choice in limb-salvageable cases and aims mainly at improving local control and reducing required surgical margins.

In head and neck MPNSTs, R0 resection may be challenging as a result of loco-regional anatomy and critical surrounding structures. Subsequently, this subgroup of patients often experiences worse oncological outcomes. In this subset of patients, the optimum choice of treatment may be surgery with adjuvant therapy [60].

Despite combined therapies and improved surgical techniques, the 5-year local recurrence rate for localized MPNST ranges from 27.3% to 85.7%, whereas median overall survival is on average 5–8 years [61].

The resection of the MPNSTs always requires the removal of a nerve, with reported rates of motor deficit up to 30% of the cases [62]. Furthermore, functional reconstructions are still not common practice, both for sensory and motor deficits [63,64,65]. Early consideration on the preservation of function, especially in the era of LPS, should be considered. Not all MPNSTs will need reconstructions, as not all of them will arise from the major nerves or require the excision of adjacent nerves, tendons, or large muscle parts. The selection of the procedure should be patient and tumour-site specific, but when large muscle resections are required, free functioning muscle transfers ought to be considered, while more distal defects may be reconstructed with the use of tendon transfers [63,66].

Nerve reconstructions are rarely performed in any soft tissue sarcomas, with only a few cases being reported in the literature. Although reconstruction may provide a good return of function, nerves regenerate slowly and often the patient requires a long rehabilitation period. As such, it is important that the life expectancy be sufficient for the expected rehabilitation period [67,68,69].

The ideal timing of reconstruction is disputed. Early reconstruction is technically less complex as there is less fibrosis, improving nerve and vessel identification and subsequently reducing potential complications [70,71,72]. As such, direct reconstruction after primary resection has shown better functional results over delayed surgical reconstruction. There is, however, some remaining concern about the potential for early reconstruction to limit further oncological management of the tumour bed in cases where close or involved surgical margins are seen.

Pre- or post-operative radiotherapy has a significant role in localized disease by reducing the risk of local relapse in MPNSTs of the extremities, trunk or head and neck, especially when there are close margins or R1 on the surgical specimen [73].

The long-term outcome of RT results in excellent local control and improved progression-free survival (PFS), but unclear benefit on survival. The usual dose of RT delivered was often in excess of 50 Gray (Gy) via different types of external RT [73,74]. The literature generally recommends that RT is most useful in patients with large (>5 cm), high-grade tumours and/or those with R1 resections [74,75]. In addition, RT timing (pre- vs. post-operative) does not appear to significantly impact the local control. Brachytherapy and intraoperative electron radiation therapy have also been explored in MPNST. According to Wong et al.’s study, the 5-year local control was 88% in patients treated with brachytherapy and 51% in those treated with external beams; therefore, the combination may be more effective [76,77,78].

There is limited available data about the role of chemotherapy in the neo-/adjuvant setting. Reported studies generally indicate a lack of survival benefit from peri-operative chemotherapy in the treatment of localized MPNST. Most of these studies were small and retrospective, incorporating patients with different chemotherapy regimens. The majority of studies have used as preferred systemic chemotherapy ifosfamide in association with an anthracycline—usually doxorubicin [79,80,81]. One Italian study that showed a survival benefit to adjuvant chemotherapy utilized epirubicin. Brunello et al. demonstrated a benefit of adjuvant chemotherapy in both disease-free (DFS) and overall survival (OS) in patients with high-risk STS (size ≥ 5 cm, high grade and stage III), including MPNST (8% of all patients). There was benefit of adjuvant chemotherapy versus untreated (median DFS 29.6 months vs. 7.8 months, HR-0.32; median OS 67 months vs. 33.7 months, HR-0.41) [82].

Two large prospective observational studies of STS, including MPNST in young patients, have assessed outcomes in relation to treatment approach. In the ARST0032 study of non-rhabodmyosarcomatous STS in patients younger than 30, MPNST was the second most represented STS subtype after synovial sarcoma (MPNST—58 patients (11%)) [83]. In this trial, newly diagnosed patients were assigned to one of the three risk groups (based on extent of surgical resection, tumour size and grade and presence/absence of distant metastasis). Patients were then allocated to one of four risk-adapted treatment strategies (A—surgery only, B—surgery + adjuvant radiotherapy (RT), C—surgery + adjuvant chemotherapy (CHT) and radiotherapy and D—neoadjuvant chemotherapy and radiotherapy, followed by surgery and then adjuvant chemotherapy and radiation therapy). Doxorubicin (37.5 mg/m^2^/dose) and ifosfamide (3 g/m^2^/dose) were utilized in both arm C and D. At a median follow-up time of 6.5 years, the study’s 5-year overall survival (OS) and event-free survival (EFS) were as follows: for low risk: 96.2% and 88.9%, respectively; intermediate risk: 79.2% and 65.0%, respectively; high risk: 35.5% and 21.2%, respectively. This trial indicated that surgery alone is an adequate treatment for lower risk patients, while intensive multimodality approaches for higher risk patients were associated with limited survival.

In the European Paediatric Soft Tissue Sarcoma Group (EpSSG) NRSTS-2005, patients younger than 21 with non-rhabdomyosarcomatous STS were stratified into 4 treatment groups based on the use or omission of peri-operative radiotherapy or chemotherapy [84]. Moreover, 51 patients with MPNST were included, of whom 26 (51%) had a background of NF1. Of these patients, 13 (25%) had surgery alone (Group 1), 4 (8%) received adjuvant radiotherapy (Group 2), 7 (14%) had adjuvant chemotherapy +/− radiotherapy (3 cycles of doxorubicin 75 mg/m^2^ + ifosfamide 9 g/m^2^ and then 2 cycles of concomitant ifosfamide) (Group 3) and 27 (53%) received neoadjuvant chemotherapy +/− radiotherapy (Group 4). In patients who received pre-operative therapy, radiological response was seen in 46%. Furthermore, 5-year event-free and overall survival across all 51 patients was 52.9% and 62.1%, respectively. The 5-year EFS was 92%, 33%, 29% and 42% for treatment groups 1–4, respectively—the superior outcomes in the surgery alone group likely reflect selection bias. Of note, however, is the numerically superior outcomes in patients who received pre-operative compared to post-operative chemo/radiotherapy.

Peri-operative chemotherapy may provide a marginal survival benefit in patients with high-grade soft tissue sarcoma. The SARC006 trial assessed an intensive schedule of ifosfamide in combination with doxorubicin and then etoposide in the neoadjuvant treatment of locally advanced or metastatic MPNST. Promising rates of disease control were seen in this trial, facilitating radical resection in a majority of patients with localized disease. There was some indication that tumour response was seen less frequently in NF1-related tumours than in sporadic cases (ORR 17% vs. 44%), but the small sample size prevented formal statistical confirmation. At present, the use of peri-operative chemotherapy should be considered on a case-by-case basis [85,86].

Adjuvant/neo-adjuvant RT is still debatable for localized MPNSTs but should be really considered in a high-grade MPNST, whereas for the majority of low-grade localized MPNSTs, surgery alone is recommended.

## 5. Local Recurrence

There is the consideration of re-resection if there is an apparent macroscopic tumour remaining/recurring in the tumour bed. The treatment of recurrences remains uncertain and varies in practice. The main objective of treatment at this stage is to prolong DFS. Therefore, it is essential to understand the impact of any management option on overall survival and the likelihood of a potential second relapse. Based on the MONACO study, 507 patients were included, and 142 of them developed a local recurrence (LR) during their follow-up time. There was a higher incidence of NF1-related MPNSTs relapses (40% vs. 31%), usually large (>5 cm, approx. 55%), and most of them were located in the extremities (>50%). Patients who develop an LR1 often have initial high-grade tumours (92.2% vs. 83.4%) and microscopically positive margins (R1) (39.4% vs. 33.2%). Patients with a LR1 were mostly treated with surgery only for their primary tumour (44.4%) or surgery and adjuvant RT (43.0%).

Predictive factors of local relapse were original high-tumour grade, R1 resection and tumour size > 5 cm. The use of RT reduced the risk of relapse. Almost two-thirds of the patients (64.9%) had surgery for their recurrence. R0 resections were achieved in more than a third of these patients, whereas up to 20% of them had no treatment. Of all patients that had a local relapse, 22.5% also had metastasis.

Surgically treated patients were associated with better OS (HR 0.38). The median OS in patients surgically treated for their LR was 56 months, compared to 43 months in patients without surgery for their LR [87,88].

## 6. Inoperable or Metastatic Disease

For inoperable or metastatic MPNST, the outcome is usually poor. The standard of care in this setting remains chemotherapy. An anthracycline-based chemotherapy (doxorubicin) single or in combination is the preferred first-line option, with response rates varying from 20 to 60%, depending on different studies [79,80,89]. Although the highest response rates are seen in regimens containing ifosfamide, there is no survival superiority to adding this drug (based on ECOG group—median OS 8.8 months in single agent doxorubicin versus 11.5 months in doxorubicin + ifosfamide) [90]. Despite initial responses, the prognosis still remains low, and these responses are rapidly followed by accelerated progression and death. In general, just 20–30% of the patients will survive two years post-diagnosis.

In terms of effectiveness, subsequent lines of chemotherapy are considerably lower than first-line treatment, reflecting the aggressive behaviour of MPNSTs in this setting. Data are lacking on the MPNST-specific efficacy of further lines of systemic therapy commonly used in advanced STS (for instance, gemcitabine in combination with Docetaxel or dacarbazine and trabectedin), but objective response rates (ORR) are seen in around 10–15% of the cases. An important aspect seen in previous studies showed that sporadic MPNST had a better response as compared to NF1-associated MPNST [91].

As with other STS subtypes, MPNST with oligometastatic disease can be approached with aggressive locoregional therapies with the objective of achieving a ‘no evidence of disease’ status and an aim of significantly altering the overall course of advanced disease. Such modalities can include surgery, high-dose ablative radiotherapy or percutaneous needle ablations. Yan et al. used microwave ablation as a salvage procedure for a patient with a large intra-abdominal MPNST that relapsed within 2 months of radical resection. Despite immediate necrosis shown on the scan studies post-ablation, the patient died in less than 3 months due to further enlargement of the tumour and rapid deterioration [92]. The evidence based to support such aggressive loco-regional management of oligometastatic disease is limited to non-comparative and generally retrospective series that do not provide direct evidence of any survival advantage compared to more conservative, systemic therapy-based approaches. Most STS guidelines support such approaches in carefully selected patients.

While an increasing amount is understood around the molecular pathology of MPNST, little progress has been made in developing molecularly targeted therapies. The PI3K/AKT/mTOR pathway has been shown to be upregulated in MPNST via constitutive RAS activation that results from neurofibromin loss-of-function [93]. In vitro studies showed that mTOR inhibition by everolimus has antitumour activity in MPNST cell lines [94]. In the SARC016 clinical trial, the combination of bevacizumab and everolimus achieved a clinical benefit rate of 12%, which was considered ineffective in the trial [94]. Currently, the SARC031 trial is evaluating combined MEK/mTOR inhibition (NCT03433183), further emphasizing the need for trials that combine inhibitors with preclinical justification in MPNST.

BRAF V600E is a novel target for MPNST therapy. There was an isolated report by Kaplan who described a female MPNST patient with a BRAF mutation who received Vemurafenib for approximately 4 days, and following this, the tumour had shrunk by 50% [49].

Finally, the immunotherapy field is starting to be investigated in the treatment of MPNSTs. There are scarce case reports showing some activity in metastatic disease [95]. Also, there are several ongoing clinical trials of immunotherapies in patients with MPNSTs, mostly checkpoint inhibition and oncolytic viruses. Upregulation of ligands like PD-L1 is common alongside MPNSTs, providing a valid point to further explore the possible benefit of the blockade of PD-1 and/or PD-L1 [96,97]. Although only a few individual case studies have been reported, each patient achieved a complete response, likely due to a PD-L1 positivity [98]. The Alliance A091401, a phase II trial evaluating the use of nivolumab alone versus the combination of nivolumab and ipilimumab in patients with advanced/metastatic soft tissue sarcoma, which included two patients with MPNST, showed an overall response rate ranging from 5 to 16%. The combination seemed to be specifically more effective in certain sarcoma subtypes (undifferentiated pleomorphic sarcoma and liposarcoma), whereas no response was seen in any of the MPNST cases. By improving the knowledge of understanding the role of checkpoint inhibitors in sarcoma cases, there is a need to develop a biomarker to determine which sarcoma patients are more likely to benefit from checkpoint blockade [99].

Given the poor outcome with available systemic treatments, clinical trials for MPNST are encouraged. There have been several trials using targeted therapies in an attempt to find more encouraging treatment options, the combination of MEK inhibitors with various combinations of immune checkpoint inhibitors and inhibitors of BRD4, MDM2 or TYK2 (Table 1). In MPNST, as described above, there are multiple epigenetic dysregulations of transcriptional factors and kinase signalling that could be targetable by using different combination drug therapies that have synergistic effects [100]. Epigenetic-based therapies have gained attention in MPNST cases over the last years. In up to 90% of all MPNST cases, loss of function mutations in SUZ12 or EED, which encode PRC2, have been identified. These mutations are associated with the malignant transformation from a benign PN to MPNST. A potential treatment strategy is to reverse the PCR2 activity loss with bromodomain inhibitors [16].

## 7. Conclusions

Considering the rarity of these tumours and the relative lack of the specific literature, further studies are needed for a better understanding and management of these tumours. NF1-associated MPNST is linked with a worse survival than sporadic MPNST, so an optimal treatment regimen may differ for these two entities [67,101]. The mechanism of sporadic MPNST has not been entirely identified, so further studies are needed to assess the risk and select an appropriate treatment strategy tailored on a potential targetable alteration.

In clinical practice, current IHC markers provide limited specificity in the diagnosis of MPNSTs, so more precise criteria and improved genetic techniques could be the future tool for detection and diagnosis.

Despite ongoing clinical trials, complete surgical resection remains the most efficient treatment option for MPNST. The value of radiotherapy and chemotherapy for survival is still debated.

Clinical trials in MPNST are encouraged as novel targeted agents might play an important role in outcome improvement. Currently, most drugs in clinical trials are based on the RAS and MAPK/MEK pathways. Nonetheless, most of these trials have failed to provide any benefit. MPNST is a complex disease with multiple genomic alterations, and the main problem for clinical research is the scarcity of the disease and a lack of comprehensive gene sequencing.

The future of MPNST may need more exhaustive analysis, such as tailoring patients based on different genetic mutations or adopting distinctive treatment strategies depending on MPNST development (sporadic/NF1-associated) [102].

## Figures and Tables

**Figure 1 cancers-16-03266-f001:**
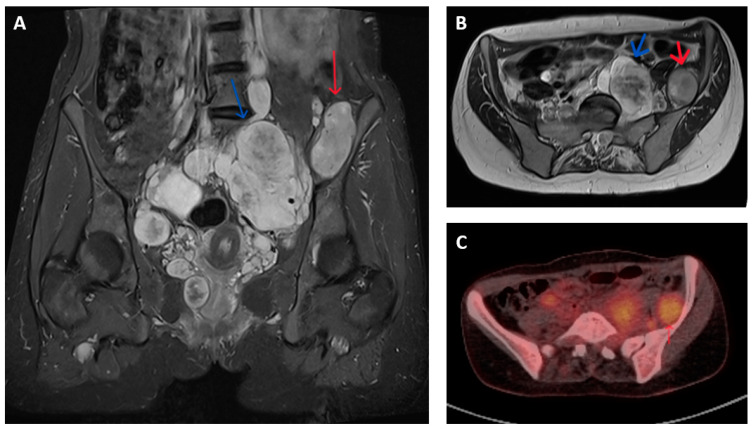
Imaging appearances of benign and malignant nerve sheath tumours in patients with NF1. T2 weighted Short Tau Inversion Recovery (STIR) sequences in (**A**) coronal and (**B**) axial planes show numerous pelvic neurofibromata (blue arrows) and a lesion adjacent to left iliac bone that had shown significant growth compared to previous imaging (red arrow). This lesion showed features of atypia but no overt malignancy on biopsy but, on subsequent resection, was confirmed as containing high-grade MPNST. (**C**) Axial FDG-PET-CT image of corresponding region of same patient shows higher grade FDG uptake (SUV max 5.7) in malignant lesion (red arrow) compared to nearby non-malignant nerve sheath tumours.

**Table 1 cancers-16-03266-t001:** Relevant active clinical trials in the MPNST population.

Trial ID	Design and Population	Agent	Molecular Targets	Trial Status
**NCT04872543**	Phase II; PRC2 loss MPNST	Cedazuridine + Decitabine	PCR2 mutation	Active, recruiting
**NCT02584647**	Phase II; any MPNST	Pexidartinib + Sirolimus	Multi-kinase inhibitor mTOR inhibitor	Active, not recruiting
**NCT02700230**	Phase I; any MPNST	MV-NIS	Oncolytic virus targeting NF1 tumour cells	Active, recruiting
**NCT04917042**	Phase II; any MPNST	Tazemetostat	EZH2 inhibitor	Active, recruiting
**NCT04465643**	Phase I; neoadjuvant pre-malignant/malignant MPNST	Nivolumab + Ipilimumab	Anti-PD1 + Anti-CTLA4 immune checkpoint inhibitors	Active, recruiting
**NCT05253131**	Phase II; any MPNST	Selumetinib + BI + Durvalumab	MAPK/MEK inhibitor + Bromodomain inhibitor + anti-PD-L1 inhibitor	Not yet recruiting
**NCT03611868**	Phase IB/II; any ≥12 years old MPNST	Alrizomadlin + Pembrolizumab	MDM2 inhibitor + anti-PD1 inhibitor	Active, recruiting

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
