# Peer review of "The Multimodality Management of Malignant Peripheral Nerve Sheath Tumours"

_cancers, 2024, doi:10.3390/cancers16193266_

Round 1

Reviewer 1 Report

Comments and Suggestions for Authors

This article reviews thoroughly yet concisely current proven evidence and emerging therapies useful for any clinician treating MPNST. Pitfalls in current literature are adequately addressed.

Comments on the Quality of English Language

Throughout the manuscript some grammatical errors are present. Please address these.

E.g. lines 206-208: Currrent guidelines recommend limb preserving surgery (LPS); nevertheless, up to 5-10% will underwent an amputation due to a more conservative approach not being feasible (59,60).

Author Response

Comment 1: Throughout the manuscript some grammatical errors are present. Please address these.

Reply to comment 1: We really appreciate your comments. After we have checked the whole manuscript, we have corrected the grammatical errors and improved sentence structures throughout our paper.

Reviewer 2 Report

Comments and Suggestions for Authors

Seres et al provide a review on the multi modal management of peripheral nerve sheath tumors. Overall, the manuscript is well written and concise. 

Major Concerns:

1) The authors should clarify if there is difference in management of high grade and low grade MPNSTs

2) The authors should clarify when biopsy should be performed for a suspected nerve sheath tumor. 

Author Response

Comment 1: The authors should clarify if there is difference in management of high grade and low grade MPNSTs.

Thank you for your insightful and constructive comments. We truly appreciate the time and effort you took to review our manuscript and provide valuable feedback. Therefore, in respect to your first comment I have to point out that the management is explained in the localised MPNST chapter where the role of neo-/adjuvant RT is usually restricted to certain MPNSTs (including high-risk ones), whereas the low risk should be usually managed with surgery alone. 

Comment 2: ) The authors should clarify when biopsy should be performed for a suspected nerve sheath tumour.

This comment is addressed in the "Clinical presentation and Diagnosis" where it is outlined that a biopsy is warranted if there are any imagistic suspicion or worsening symptomatology especially in a patient with known PN where the diagnosis might be more challenging. Once again, we thank you for your valuable feedback to our manuscript.

Reviewer 3 Report

Comments and Suggestions for Authors

This article systematically reviews the multidisciplinary treatment approaches for MPNSTs, with a particular focus on the current state of treatment for NF1-associated MPNSTs.

-The research data cited in the article are relatively limited, and it is recommended to explore more innovative aspects, such as the potential roles of new targeted therapies and immunotherapies.

-The section on clinical presentation and diagnosis is relatively lengthy. To improve the reading experience, it is suggested to shorten this part.

-Some of the cited references are somewhat outdated, especially in the rapidly evolving fields of molecular biology and targeted therapy. It is recommended to incorporate more recent research findings.

Comments on the Quality of English Language

-Some sentences in the article appear to be overly long, so simplifying complex sentence structures is advised to enhance readability.

Author Response

Comment 1: The research data cited in the article are relatively limited, and it is recommended to explore more innovative aspects, such as the potential roles of new targeted therapies and immunotherapies.

Thank you for pertinent remark. Looking into the literature, despite the scarce data regarding patients with MPNST, we were able to address this and add some additional data in respect to the novel therapeutic approaches that might change the MPNST management in the future. Therefore, in the metastatic MPNST section we have included relevant information to address this. 

Comment 2: The section on clinical presentation and diagnosis is relatively lengthy. To improve the reading experience, it is suggested to shorten this part.

Thank you for your thoughtful observation regarding the length of clinical presentation and diagnosis chapter. It is important to maintain clarity and conciseness to enhance the reader’s experience. Therefore, we have condensed less critical parts and removed any redundant information, so overall we think we were able to reduce this chapter in accordance with your request. 

Comment 3:  Some of the cited references are somewhat outdated, especially in the rapidly evolving fields of molecular biology and targeted therapy. It is recommended to incorporate more recent research findings.

Thank you for highlighting the importance of using up-to-date data. We understand the concern regarding the relevance of data in ensuring accurate and meaningful conclusions. While the data may seem outdated, it was chosen because it covers a critical period that we thought is still relevant. To clarify this point, we have added some recent research data available in the literature.